# PoseCheck: Generative Models for 3D Structure-based Drug Design Produce Unrealistic Poses

## Abstract

Deep generative models for structure-based drug design (SBDD), where molecule generation is conditioned on a 3D protein pocket, have received considerable interest in recent years. These methods offer the promise of higher-quality molecule generation by explicitly modelling the 3D interaction between a potential drug and a protein receptor. However, previous work has primarily focused on the quality of the generated molecules themselves, with limited evaluation of the 3D *poses* that these methods produce, with most work simply discarding the generated pose and only reporting a "corrected" pose after redocking with traditional methods. Little is known about whether generated molecules satisfy known physical constraints for binding and the extent to which redocking alters the generated interactions. We introduce PoseCheck, an extensive benchmarking suite for state-of-the-art SBDD methods and find that generated molecules have significantly more physical violations and fewer key interactions compared to baselines, calling into question the implicit assumption that providing rich 3D structure information improves molecule complementarity. We make recommendations for future research tackling identified failure modes and hope our benchmark will serve as a springboard for future SBDD generative modelling work to have a real-world impact.

## 1 Introduction

Structure-based drug design (SBDD) Blundell (1996); Ferreira et al. (2015); Anderson (2003) is a cornerstone of drug discovery. It uses the 3D structures of target proteins as a guide to designing small molecule therapeutics. The intricate atomic interactions between proteins and their ligands shed light on the molecular motifs influencing binding affinity, selectivity, and drug-like properties. Employing computational methods such as molecular docking Trott & Olson (2010); Alhossary et al. (2015), molecular dynamics simulations Klepeis et al. (2009), and free energy calculations Chipot & Pohorille (2007), SBDD aids in the identification and optimization of potential drug candidates.

Deep generative models for SBDD have recently attracted considerable attention in the ML community Du et al. (2022); Isert et al. (2023). These models learn from vast compound databases to generate novel chemical structures with drug-like properties Gómez-Bombarelli et al. (2018). By explicitly integrating protein structure information, these models aim to generate ligands with a higher likelihood of binding to the target protein. In particular, advancements in geometric deep learning Bronstein et al. (2017); Atz et al. (2021); Jing et al. (2020) have led to a new suite of generative methods, enabling the design of 3D molecules directly within the confines of the target protein Masuda et al. (2020); Luo et al. (2021); Peng et al. (2022); Guan et al. (2023a); Schneuing et al. (2022). These methods, which concurrently generate a molecular graph and 3D coordinates, provide the significant advantage of obviating the need for determining the 3D pose *post hoc* through traditionally slow molecular docking programs – at least in theory.

Assessing the quality of molecules generated by these methodologies is not straightforward, with little work on experimental validation, especially for *de novo* design Baillif et al. (2023). Typical evaluation metrics (Figure 1a) focus primarily on the 2D graph of the generated molecules themselves, measuring their physicochemical properties (e.g. QED Bickerton et al. (2012)) and adherence to drug discovery heuristics (e.g. Lipinski's Rule of Five Lipinski et al. (2012)). For effective

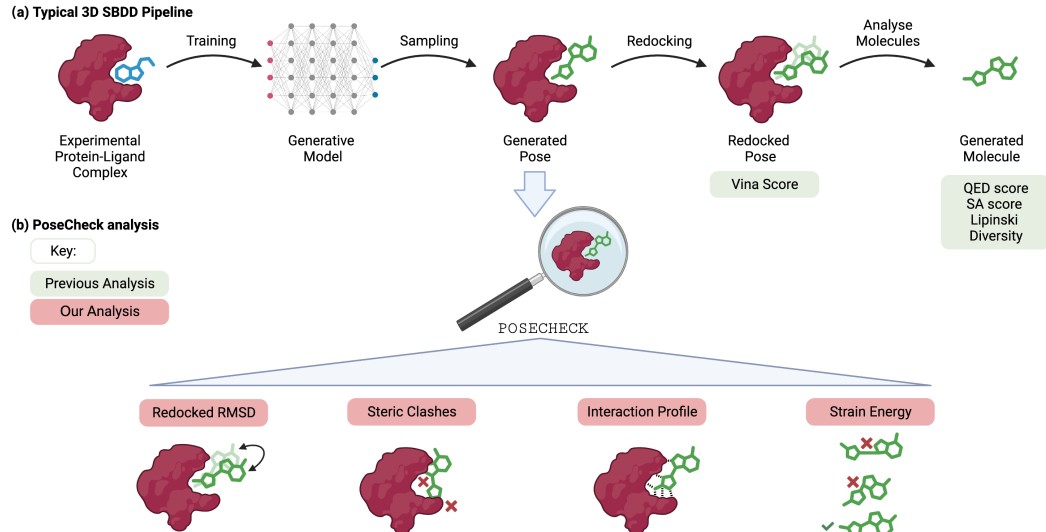

Figure 1: **Top:** Overview of a conventional pipeline of SBDD with 3D generative modelling. A generative model is usually trained using experimental or synthetic protein-ligand complexes, from which new molecules and poses can be sampled *de novo*. Typically, generated poses are discarded and redocked into the receptor, and primarily evaluated on 2D molecular graphs (e.g. QED). The effect of redocking on the final complex is often unknown, preventing understanding of the common failure modes of the trained model and therefore inhibiting progress. **Bottom:** The POSECHECK benchmarks for generated poses include pipeline-wide as well as component-wise metrics, enabling a targeted evaluation of each model component guiding further model development.

SBDD, we argue that it's equally essential to assess the quality of the generated *binding poses* and their capacity to satisfy known biophysical prerequisites for binding (Figure 1b). This perspective is essential if these methods are to serve as practical alternatives to traditional virtual screening approaches in SBDD.

We hypothesise that multiple failure modes, undetected by currently applied metrics, are inherent within these methods. The situation is further complicated by the common practice of disregarding the initially generated pose and then redocking the molecule to attain a potentially enhanced pose and only reporting these scores Peng et al. (2022); Luo et al. (2021); Schneuing et al. (2022); Zhang et al. (2022).

Our primary contributions are summarized as follows:

1. We introduce POSECHECK, a set of new biophysical benchmarks for SBDD models, expanding the traditional 'pipeline-wide' framework by integrating 'component-wise' metrics (i.e. generated and redocked poses), leading to comprehensive and precise model assessment.

2. Utilizing this new framework, we evaluate a selection of high-performing machine learning SBDD methods, revealing two key findings: (A) generated molecules and poses often contain nonphysical features such as steric clashes, hydrogen placement issues, and high strain energies, and (B) redocking masks many of these failure modes.

3. Based on these evaluations, we propose targeted recommendations to rectify the identified shortcomings. Our work thus provides a roadmap for addressing critical issues in SBDD generative modelling, informing future research efforts.

## 2 BACKGROUND

**Deep Generative Models for 3D Structure-based Drug Design** Many works have recently tried to recast the SBDD problem as learning the 3D conditional probability of generating molecules given

a receptor, allowing users to sample new molecules completely *de novo* inside a pocket. Common methods utilize Variational AutoEncoders (VAEs) Kingma & Welling (2013), Generative Adversarial Networks (GANs) Goodfellow et al. (2014), Autoregressive (AR) models and recently Denoising Diffusion Probabilistic Models (DDPMs) Ho et al. (2020). LiGAN Masuda et al. (2020) uses a 3D convolutional neural network combined with a VAE model and GAN-style training. 3DSBDD Luo et al. (2021) introduced an autoregressive (AR) model that iteratively samples from an atom probability field (parameterised by a Graph Neural Network) to construct a whole molecule, with an auxiliary network deciding when to terminate generation. Pocket2Mol Peng et al. (2022) extended this work with a more efficient sampling algorithm and better encoder. DiffSBDD Guan et al. (2023a), DiffBP Lin et al. (2022) and TargetDiff Guan et al. (2023a) are all conditional DDPMs conditioned on the 3D target structure. DecompDiff Guan et al. (2023b) is another diffusion model that decomposes the ligand into fragments for which it considers separate priors for the diffusion process. FLAG Zhang et al. (2022) chooses a fragment from a motif vocabulary based on the protein structure and composes it with other motifs into a final ligand in an iterative fashion. GraphBP Liu et al. (2022) utilises an autoregressive flow model to formulate the ligand design as a sequential generation task.

**Related work** Guan et al. (2023a) perform limited analysis of small chemical sub-features, such as agreement to experimental atom-atom distances and the correctness of aromatic rings within the generated molecule. Baillif et al. (2023) emphasize the necessity of 3D benchmarks for 3D generative models. However, both of these works study the molecules in isolation rather than the protein-ligand context. Both DecompDiff Guan et al. (2023b) and DiffBP Lin et al. (2022) take steric clashes into account via their loss functions, but do not include steric clashes as a metric in their evaluation. TargetDiff Guan et al. (2023a) includes an analysis of Vina Scores but does not report any standard deviations on these. However, these standard deviations are critical in evaluating the performance of these models as we demonstrate in this paper. The concurrent work PoseBusters Buttenschoen et al. (2023) also focuses on benchmarking the biophysical plausibility of protein-ligand poses but focuses on evaluating *docking tools* instead of molecular generation models. They also find generalisation to new sequences to be poor.

## 3 METHODS

In order to evaluate the quality of generated poses and their capacity to facilitate high-affinity protein-ligand interactions, we present a variety of computational methods and benchmarks in this section. These methodologies provide a thorough perspective on the poses produced and illuminate the ability of generative models to generate trustworthy and significant ligand conformations. Full implementation details are given in Appendix A.

**Interaction fingerprinting** Interaction fingerprinting is a computational method utilized in SBDD to represent and analyze the interactions between a ligand and its target protein. This approach encodes specific molecular interactions, such as hydrogen bonding and hydrophobic contacts, in a compact and easily comparable format – typically as a bit vector, known as a *interaction fingerprint* Bouysset & Fiorucci (2021); Marcou & Rognan (2007). Each element in the interaction fingerprint corresponds to a particular type of interaction between the ligand and a specific residue within the protein binding pocket. We compute interactions using the ProLIF library Bouysset & Fiorucci (2021).

**Steric clashes** In the context of molecular interactions, the term *steric clash* is used when two neutral atoms come into closer proximity than the combined extent of their van der Waals radii Ramachandran et al. (2011). This event indicates an energetically unfavourable Buonfiglio et al. (2015), and thus physically implausible, interaction. The presence of such a clash often points towards the current conformation of the ligand within the protein being less than optimal, suggesting possible inadequacies in the pose design or a fundamental incompatibility in the overall molecular topology. Hence, the total number of clashes serves as a vital performance metric in the realm of SBDD. We stipulate a clash to occur when the pairwise distance between a protein and ligand atom falls below the sum of their van der Waals radii, allowing a clash tolerance of 0.5 Å.

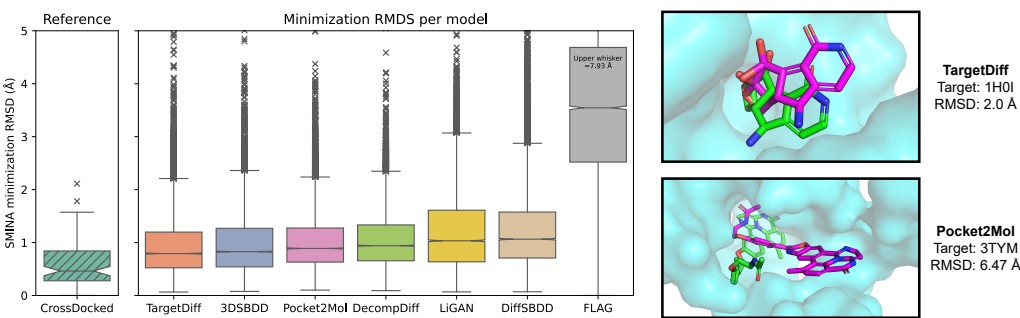

Figure 2: **Left:** RSMD between the generated and SMINA minimized poses for CrossDocked and all generative methods (note FLAG upper whisker value is not shown to preserve a meaningful scale). **Right:** Examples of large conformational rearrangements in the ligand upon redocking.

**Strain-energy**   Strain energy refers to the internal energy stored within a ligand as a result of conformational changes upon binding. When a ligand binds to a protein, both the ligand and the protein may undergo conformational adjustments to accommodate each other, leading to changes in their bond lengths, bond angles, and torsional angles. These changes can cause strain within the molecules, which can affect the overall binding affinity and stability of the protein-ligand complex Perola & Charifson (2004). Whilst there is always a trade-off between enthalpy and entropy, generally speaking, lower strain energy results in more favourable binding interactions and potentially more effective therapeutics. We calculate the strain energy, $E_{\text{strain}} = E_{\text{generated}} - E_{\text{minimum}}$ as the difference between the estimated energy minimum and the energy of the generated pose (without pocket). Note that evaluating the generated poses with the force-field directly will cause the energy terms to explode, due to the slight imperfections in bond distances and angles in the generated molecules. Hence, we first perform at most 200 steps of relaxation using a force-field with a maximum allowed displacement in atom positions of 0.1 Å. This fixes minor issues with the bond angles and distances, preventing the energy terms from exploding, whilst staying faithful to the global binding mode of the generated conformer. An estimate of the global energy minimum is calculated by initializing 50 conformers using ETKDGv3 and then minimizing using up to 200 steps of force-field minimization (taking inspiration from Buttenschoen et al. (2023)). We then calculate the energy of all these poses and take the minimum as our final value. Both conformer minimization and energy evaluation are computed using the Universal Force Field (UFF) Rappé et al. (1992) using RDKit.

**Docking**   Our final assessment involves measuring the level of agreement between the docking programs and the molecules produced by the learned distribution in the generative model. Although this is the most coarse-grained approach we employ and docking programs come with their inherent limitations, they nevertheless contain useful proxies and serve as valuable tools for comparison. In this procedure, we redock the generated pose using SMINA Koes et al. (2013). Next, we compute the Root Mean Squared Deviation (RMSD) between the generated pose and the docking-predicted one across all generated molecules, thereby obtaining a distribution of RMSD values.

## 4 RESULTS

### 4.1 EXPERIMENTAL SETUP

In our study, we evaluate the quality of poses from seven recent methods: LiGAN Masuda et al. (2020), 3DSBDD Luo et al. (2021), Pocket2Mol Peng et al. (2022), TargetDiff Guan et al. (2023a), DiffSBDD Schneuing et al. (2022), DecompDiff Guan et al. (2023b) and FLAG Zhang et al. (2022). All models were trained on the CrossDocked2020 Francoeur et al. (2020) dataset using the dataset splits computed in Peng et al. (2022), which used a train/test split of 30% sequence identity to give a test set of 100 target protein-ligand complexes which we use for evaluation. For each model, we sampled 100 molecules per target. We give a more detailed overview of the CrossDocked dataset and its limitations in Appendix A.

During inference, the model is given a reduced PDB file containing only the atoms for a single pocket within the test set, so there is no element of blind docking during generation or subsequent redocking[1].Docking protocols were done using the SMINA settings described in the original Cross-Docked paper Francoeur et al. (2020).

## 4.2 AGREEMENT WITH DOCKING SCORING FUNCTIONS

**Results**  To discern whether the generated poses/binding modes produced by these models correspond to overall low energy states with few physical violations, our preliminary analysis involves determining the extent to which minimized poses preserve information from the initially generated binding mode. Therefore, we proceed to compute the RMSD between the model-generated pose and SMINA-minimized pose Koes et al. (2013), with a lower RMSD value denoting a higher degree of agreement.[2]

The distributions of SMINA-minimization RMSDs of various methods are illustrated in Figure 2. We first consider CrossDocked as a baseline, which has a mean minimization RMSD of 0.59 Å. Given that all the generative models were trained on these poses, we would expect to observe similar performance. However, we find that all methods (except FLAG) have a mean score between 0.94 and 1.28 Å, suggesting that the generated binding poses are very far from low-energy states. We observe little correlation between method types here except for the two similar autoregressive models, 3DSBDD and Pocket2Mol, which obtain mean RMSDs of 0.99 and 1.02 Å respectively. FLAG is the most egregious example with on average 3.64 Å RMSD during minimization and a maximum value of 10.72 Å, an extreme value for local minimisation.

We also provide the raw affinities from our docking experiments in Table 1, both when evaluating the generated pose using the SMINA/Vina score function directly (Vina Generated), after local energy minimization (Vina Minimize) and redocking the molecule entirely (Vina Redock). Additionally, we provide the change in Vina scores during minimization and redocking. We first draw our attention to the scores for the redocked poses: these metrics are commonly reported and often used as a justification for state-of-the-art performance. On the surface, the results look promising, with most methods matching or exceeding the performance of the baseline dataset (although with no statistical significance). However, a worrying picture emerges when we measure the generated poses directly, with none of the models (except LiGAN) outperforming the baseline dataset. The mean scores for CrossDocked are -5.50 kcal/mol, whereas the generative models (except LiGAN and FLAG) have mean scores between -1.94 and -5.36 kcal/mol. FLAG again performs poorly with the Vina scores for generated poses exploding to +94.20 kcal/mol, suggesting the generated poses are highly implausible.

We next consider the role energy minimization and redocking have on these final scores by considering the change in affinity during the two processes respectively. The result of this analysis highlights that minimization/redocking is critical to getting acceptable scores out of these methods, calling into question the reliability of the generated poses. CrossDocked has a Δaffinity minimization score of -0.74 kcal/mol, whereas the generative models (excluding LiGAN and FLAG) have scores between -0.75 and -3.91 kcal/mol. FLAG has a score of -89.31 kcal/mol (unsurprising given the generated poses). We see a similar picture for the impact of redocking, where the majority of methods see substantially greater increases in their scores during the procedure. In conclusion, we show that only reporting the mean Vina score of redocked poses hides critical failure modes found in many models. [3]

**Discussion**  These findings raise concerns for several reasons. They expose the minimal concordance between the binding models learned by these methods and the established SMINA methodology Alhossary et al. (2015), despite it being the source of training data. More critically, they underline the lack of accurate evaluations of generative models' capability to produce realistic binding poses; instead, these models tend to generate drug-like molecules with vague binding modes, later rectified through docking.

---

[1]Note illustrative figures may show full proteins.

[2]To provide perspective, it's worth noting that a carbon-carbon bond generally measures 1.54 Å in length.

[3]Given the acceptable redocked scores of FLAG we do not believe we have made an error in training.

Table 1: Vina score values of generated poses, energy minimized poses and redocked poses. We additionally provide the change in Vina score during minimization and redocking respectively.

| Method | Vina Generated (↓) (kcal/mol) | Vina Minimized (↓) (kcal/mol) | Vina Redocked (↓) (kcal/mol) | ∆Affinity Minimization (↑) (kcal/mol) | ∆Affinity Redocking (↑) (kcal/mol) |
|---|---|---|---|---|---|
| CrossDocked | $-5.50 \pm 2.86$ | $-6.24 \pm 2.52$ | $-6.86 \pm 2.37$ | $-0.74 \pm 1.24$ | $-1.37 \pm 1.43$ |
| TargetDiff | $-5.36 \pm 3.79$ | $-6.72 \pm 2.83$ | $-7.35 \pm 2.51$ | $-1.35 \pm 1.99$ | $-1.99 \pm 2.59$ |
| 3DSBDD | $-5.04 \pm 2.58$ | $-5.85 \pm 2.42$ | $-6.29 \pm 2.22$ | $-0.80 \pm 0.76$ | $-1.25 \pm 1.0$ |
| Pocket2Mol | $-4.55 \pm 3.18$ | $-6.38 \pm 2.92$ | $-6.96 \pm 2.72$ | $-1.83 \pm 1.66$ | $-2.40 \pm 2.01$ |
| DecompDiff | $-4.25 \pm 3.16$ | $-5.91 \pm 2.14$ | $-6.56 \pm 2.03$ | $-1.66 \pm 2.39$ | $-2.31 \pm 2.69$ |
| LiGAN | $-6.03 \pm 2.83$ | $-6.78 \pm 2.71$ | $-7.36 \pm 2.56$ | $-0.75 \pm 0.78$ | $-1.34 \pm 1.09$ |
| DiffSBDD | $-1.94 \pm 10.31$ | $-5.85 \pm 3.19$ | $-7.00 \pm 2.01$ | $-3.91 \pm 8.62$ | $-5.07 \pm 9.92$ |
| FLAG | $94.20 \pm 89.46$ | $4.89 \pm 19.36$ | $-5.69 \pm 4.19$ | $-89.31 \pm 78.45$ | $-99.89 \pm 88.94$ |

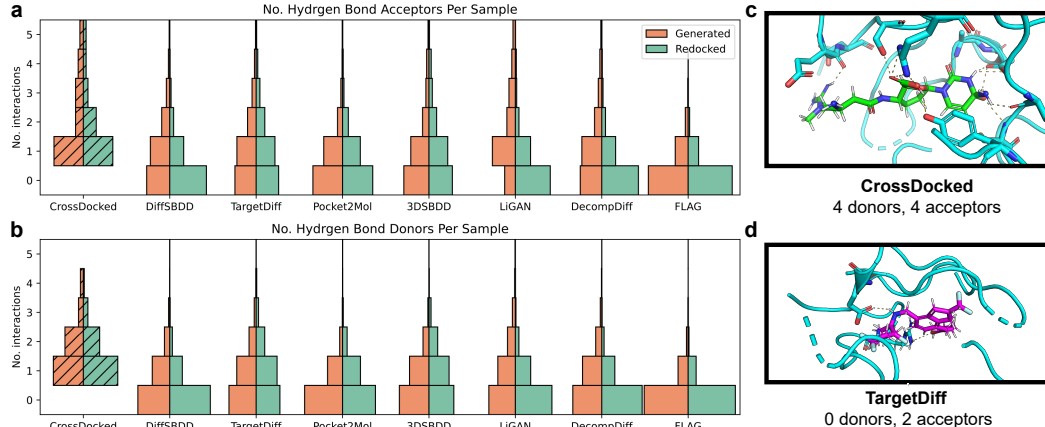

Figure 3: Interactions between protein and ligands as seen in generated poses (orange) and redocked poses (green). The frequency of (a) hydrogen bond acceptors and (b) hydrogen bond donors are considered. We find that generative models have significant trouble making hydrogen bond interactions compared to baseline (shaded boxes). Vertical histogram box sizes are normalised along the x-axis such that all have the same area. (c) Example from CrossDocked with large hydrogen bonding network. (d) Typical example from a generative models with low number of HBs.

We also calculated the RMSD between the generated and highest affinity redocked pose but were not able to discern any reasonable signal-to-noise over the baseline dataset. We hypothesise that this may be due to the fact that Francoeur et al. (2020) provided up to 20 poses for every ligand, resulting in 22.5 million complexes, and the processing done in Peng et al. (2022) is not clear on which poses they chose, meaning these models may not have been trained on the lowest affinity poses.

## 4.3 PROTEIN-LIGAND INTERACTION ANALYSIS

**Evaluation** Below describe the classes of interaction that we evaluate. **Hydrogen bonds** (HBs) are a type of interaction that occurs between a hydrogen atom that is bonded to a highly electronegative atom, such as nitrogen, oxygen, or fluorine Pimentel & McClellan (1971). They are key to many protein-ligand interactions Chen et al. (2016) and require very specific geometries to be formed Brown (1976). The directionality of HBs confers unique identities upon the participating atoms: hydrogen atoms attached to electronegative elements are deemed 'donors', whilst the atom accepting the HB is termed an 'acceptor'. **Van der Waals contacts** (vdWs) are interactions that occur between atoms that are not bonded to each other. These forces can be attractive or repulsive and are typically quite weak Andersson et al. (1998). However, they can be significant when many atoms are involved, as is typical in protein-ligand binding Barratt et al. (2005). **Hydrophobic interactions** are non-covalent interactions that occur between non-polar molecules or parts of molecules in a water-based environment. They are driven by the tendency of water molecules to form hydrogen bonds with each other, which leads to the exclusion of non-polar substances. This exclusion principle prompts these non-polar regions to orient away from the aqueous environment and towards each other Meyer et al. (2006), thereby facilitating the association between protein and ligand molecules Patil et al. (2010).

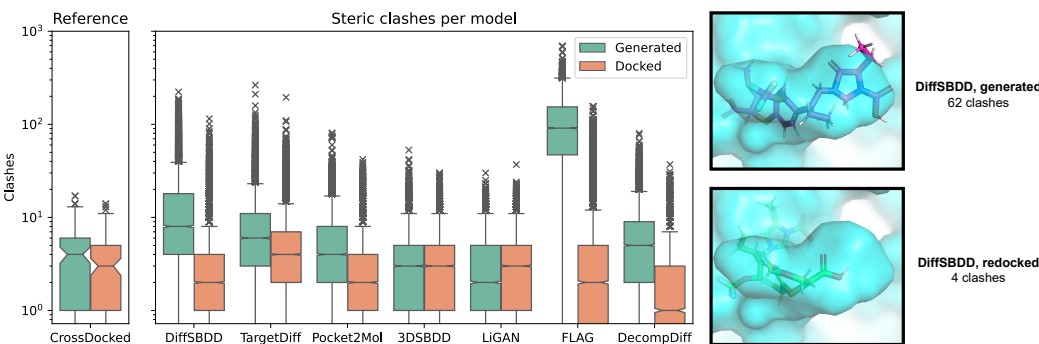

Figure 4: **Left:** number of steric clashes for the CrossDocked reference dataset as well as for the molecules generated by each model, both before and after redocking. **Right:** examples of a generated pose (magenta) and the same pose after redocking (green).

**Results** Distributions of hydrogen bonding interactions are shown in Figure 3. We consider whether our generative models can design molecules with adequate hydrogen bonding and find that no method can match or exceed the baseline. In the reference set, CrossDocked, the modal number of HBs for both acceptors and donors is 1, with means of 2.23 and 1.66 for acceptors and donors respectively. Strikingly, we find that in all generated poses for all models (except LiGAN HB acceptors) the *most common number of HB acceptors and donors is 0*, with means varying between 0.36-1.73 for HB acceptors and 0.26-0.85 for HB donors. We find an average difference of 0.50 and 0.81 HBs between the best-performing models and the baseline for acceptors and donors respectively. Results for Van der Waals contacts and hydrophobic interactions are closer to the dataset baseline (see Appendix Figure 6), possibly as these are easier to form.

**Discussion** Conventional wisdom would suggest that many minor imperfections in the generated pose would be simply fixed by redocking the molecule (e.g. moving an oxygen atom slightly to complete a hydrogen bond.) We find this is in fact rarely the case, with redocking sometimes being significantly deleterious (see examples of LiGAN in Figure 3), suggesting that there are either limitations in the docking function used or, more likely, the generated interaction was physically implausible to begin with.

## 4.4 Clash scores

**Results** Figure 4 presents the results of the steric clash analysis. In summary, the latest methods, particularly those employing diffusion models and fragment libraries, exhibit poor performance in terms of steric clashes compared to the baseline, with a significant number of outliers. Although redocking mitigates clashes to some extent, it does not always resolve the most severe cases.

The CrossDocked test set has a low number of clashes with few extreme examples, with a mean of 4.59, upper quantile of 6 and maximum value of 17. In terms of generated poses, the older methods perform best, with 3DSBDD and LiGAN having means of 3.79 and 3.40 clashes respectively. Pocket2Mol, an extension of 3DSBDD, performs worse with a mean clash score of 5.62 and upper quantile of 8 clashes. Finally, the diffusion-based approaches perform poorly with mean clash scores of 15.33, 9.03 and 7.13 for DiffSBDD, TargetDiff and DecompDiff respectively. The tail end of their distributions is also high, with the methods having upper quantiles of 18, 11 and 9 clashes respectively, with TargetDiff having the worst case of 264 steric clashes. FLAG has the worst generated clash scores by far, with mean and median clash scores of 110.96 and 91 respectively. Redocking the molecules generally fixed many clashes and improved scores, especially for FLAG, where the mean clash score improves from 110.96 to 5.55. The mean clash score for Pocket2Mol improves from 5.62 to 2.98, TargetDiff from 9.08 to 5.79 and DiffSBDD from 15.34 to 3.61.

**Discussion** Interestingly, DiffSBDD and TargetDiff, which are considered state-of-the-art based on mean docking score evaluations Guan et al. (2023a); Schneuing et al. (2022), exhibit subpar performance in their number of clashes. They aim to learn atom position distributions without

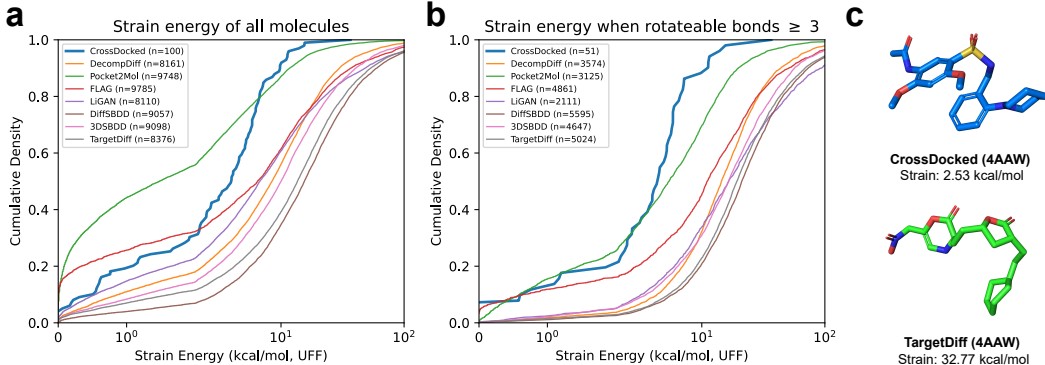

Figure 5: (a) CDF of strain energies for all molecules. (b) CDF of strain energy for all molecules that 3 or more rotatable bonds. **All of the generated molecules with 3 or more rotatable bonds are more strained than the baseline**. (c) Example of TargetDiff generated molecule with strain substantially higher than in the CrossDocked test set (target `4AAW`).

explicit constraints on final placements. While DiffSBDD starts with a performance deficit, its enhanced clash mitigation during redocking elevates its results to match the baseline, highlighting methodological distinctions between it and TargetDiff. Notably, 3DSBDD and LiGAN show low clash scores, with the former positioning atoms within a predefined voxel grid Luo et al. (2021) and the latter applying a clash loss Masuda et al. (2020). DecompDiff also applies a steric clash loss (but does not directly measure clashes in the corresponding publication) Guan et al. (2023b) and performs best out of all the diffusion-based approaches. Generated molecules for FLAG were most egregious here; we speculate this is a result of first choosing a fragment from a fragment vocabulary using a softmax function and then forcing the placement of the fragment Zhang et al. (2022), regardless of whether it fits sterically.

Our findings affirm the assumption that redocking alleviates many minor clashes, akin to the force-field relaxation step in AlphaFold2 Jumper et al. (2021). We initially speculated that molecules with clashes exceeding 100 had been mistakenly generated inside the protein pocket. Yet, we often discovered fragments within highly constrained nooks, especially worsened with the addition of hydrogen atoms.

**Limitations** An important consideration to bear in mind is that proteins are not entirely rigid receptors. They can often experience limited conformational rearrangements to accommodate molecules of varying shapes and sizes Davis & Teague (1999). Consequently, conducting generation and redocking in a rigid receptor environment may not yield accurate scores for potentially plausible molecules. Note all these results are with a *generous* clash tolerance of 0.5 Å (roughly half the vdW radii of a hydrogen atom), in order to be able to resolve differences between methods.

### 4.5 Strain energy

To conclude our study, we provide an analysis of the strain energy Perola & Charifson (2004) of the generated poses. Force field relaxation of the generated pose before docking is a common post-processing step of many generative SBDD pipelines, masking potential issues with the precise geometries of the generated molecules. Furthermore, high strain energy would be indicative that molecules are unlikely to bind.

In Figure 5a, we present the cumulative density function (CDF) of strain energy for all molecules generated by various models, using the CrossDocked dataset as a comparative baseline. Note that the x-axis is logarithmic. The data shows that most generative methods fall short of the CrossDocked's median strain energy of 3.96 kcal/mol, with FLAG and Pocket2Mol being the significant exceptions.

Upon examining the impact of rotatable bonds on strain energy, two key findings emerged: Firstly, molecules from generative models with a high count of rotatable bonds exhibit notably higher strain (refer to Appendix Figure 7). Secondly, the more successful methods in Figure 5a tend to produce molecules with fewer or no rotatable bonds, as detailed in Appendix Figure 8.

When focusing solely on molecules with three or more rotatable bonds, none of the methods surpassed the baseline performance (see Figure 5b). This discrepancy becomes more pronounced when assessing molecules with higher conformational complexity.

## 5 RECOMMENDATIONS FOR FUTURE WORK

**Exploring reduced-noise sampling strategies**   Interestingly, both diffusion-based works (DiffS-BDD and TargetDiff) performed similarly in terms of strain energy (see Section 4.5). We hypothesize this may be due to the injection of random noise into the coordinate features at all but the last step of stochastic gradient Langevin dynamics samplings Welling & Teh (2011), making it challenging to construct precise bond angles etc. Here, inspiration could be taken from protein design. For example, Chroma develops a low-temperature sampling regime to reduce the effect of noise Ingraham et al. (2022), FrameDiff effectively scales down injected noise Yim et al. (2023), both resulting in a substantial increase in sample quality with an acceptable decrease in sample diversity.

**Penalising steric clashes during training**   All evaluated methods frequently create steric clashes, resulting in physically unrealizable samples. We suggest that mitigating steric clashes is key for the next generation of SBDD models. This could be done via extra loss terms, for example, by including a distogram loss as in AlphaFold2 Jumper et al. (2021) or the steric clash loss in LiGAN Masuda et al. (2020) and DecompDiff Guan et al. (2023b) (note that the latter method does not explicitly measure clashes). A similar loss-based approach has been effective in mitigating chain-breaks in diffusion models for protein backbone design Yim et al. (2023).

**Consider explicit representation of hydrogens**   Virtually all work in ML for structural biology chooses to not explicitly represent hydrogen atoms Jumper et al. (2021); Peng et al. (2022); Luo et al. (2021); Yim et al. (2023); Schneuing et al. (2022); Guan et al. (2023a), under the assumption that they can be *implicitly* learned and reasoned over with deep neural networks. However, our analysis of hydrogen bond networks within generated molecules found that generative methods struggle to handle the precise geometries required to make a hydrogen bond Brown (1976) (even when redocked). Despite the increased computational cost, we therefore recommend that future work explores their inclusion.

## 6 CONCLUSION

In conclusion, this work presents a comprehensive exploration of structure-based drug design (SBDD) methodologies with deep generative models. We advocate for the need to consider *both* the quality of the generated molecules *and* the quality of the binding poses in these models, calling for an expanded evaluation of SBDD. The application of deep generative models in SBDD holds promise for developing innovative drug-like molecules. However, for SBDD approaches to realise that potential, we must establish a rigorous evaluation regimen of both the generated molecules and their interaction with the target – as proposed in this paper. Our research provides a solid evaluation regimen for future advancements in this field and we hope that it stimulates further development towards more efficient drug discovery processes. Code is available at anonymous.4open.science/r/posecheck-CFB1

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

## A CROSSDOCKED DATASET

The CrossDocked dataset is a standard dataset used in the field of generative modelling for structure-based drug design Francoeur et al. (2020); since the models benchmarked here were trained on this dataset, it is the benchmarking dataset of choice. It was originally created by clustering PDB structures by "pocket similarity' via Pocketome Kufareva et al. (2012), i.e. grouping structures with similar ligand binding sites together. To expand the dataset beyond this initial data, all ligands with a molecular weight ¡ 1000 Da that were associated with a given pocked were docked into each receptor assigned to that pocket via the docking tool smina Koes et al. (2013). This cross-docking process results in the basis dataset CrossDocked 2020 Francoeur et al. (2020), which contains 2,922 pockets, 18,450 complexes and 13,839 ligands, together comprising around 22.5 million poses (i.e. protein-ligand structures).

Most generative models are however not trained on this raw dataset, but on a filtered version of it, following the procedure of the Pocket2Mol model Peng et al. (2022). As a quality control, data points whose binding pose RMSD is greater than 1 $\mathring{A}$ were filtered out. This leads to a filtered dataset with 184,057 data points. The mmseq2 program Steinegger & Söding (2017) was used to cluster data at 30% identity, and training and test sets were created by randomly drawing 100,000 protein-ligand pairs for training and 100 proteins from the remaining clusters for testing.

The 100 proteins comprising the test set are on average around 320 residues long, with the biggest protein having a length of 752 residues.

## B EXTENDED IMPLEMENTATION

### B.1 METHODS IMPLEMENTATION

All generative methods accessed were trained using the same dataset and splits as proposed in Peng et al. (2022). Docking protocols were done using the SMINA settings decribed in the original CrossDocked paper Francoeur et al. (2020).

### B.2 PROCEDURE OF MODEL REPRODUCTION

For generated poses, we sourced molecules from Schneuing et al. (2022) for DiffSBDD, and Guan et al. (2023a) for CrossDocked, TargetDiff, Pocket2Mol, 3DSBDD and LiGAN (where they provide generated poses but we additionally perform our own redocking).

For FLAG Zhang et al. (2022), no weights were provided so we retrained the model as described in Zhang et al. (2022) using the code and config file available at `github.com/zaixizhang/FLAG`. When sampling, we found that generation was attempted 100 times per target and then any molecules with fewer than 8 atoms were discarded. This ended up encompassing the majority of molecules, resulting in small test sizes, so we implemented a while loop to sample 100 molecules whilst keeping faithful to the filtering used in the codebase. Having modified the code to work on GPU, sampling 100 targets took about 1-2 minutes per target on a single A100 GPU.

For DecompDiff Guan et al. (2023b), we use the official implementation with the published weights available at `github.com/bytedance/DecompDiff`. We sampled 100 samples for each of the 100 targets using the `sample_diffusion_drift.py` script in `ref_prior` mode. With the provided code, sampling 100 targets took about 20-30 minutes per target on a single A100 GPU.

## C ADDITIONAL FIGURES

### C.1 INTERACTIONS ANALYSIS

We include the comparisons between generative method against baselines for both Van der Waals contacts and hydrophobic interactions, both for generated redocked poses in Figure 6.

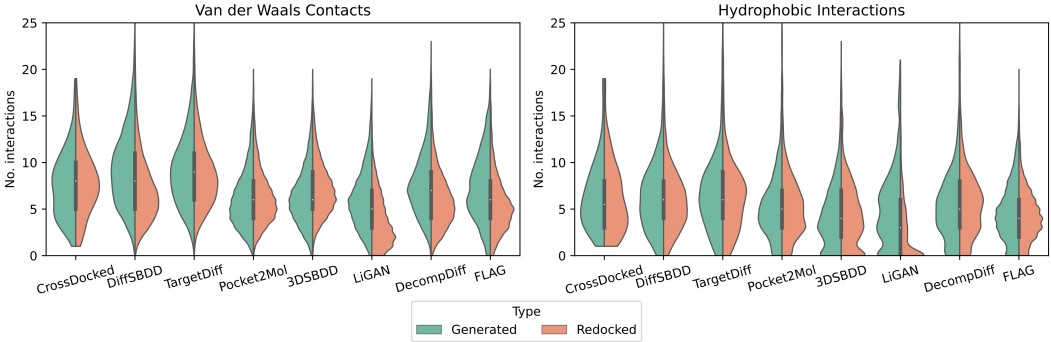

Figure 6: Extended analysis of the interaction profiles of the generated molecules for the different methods. While the focus in the main text was on hydrogen bonds, the results in this figure include Van der Waals Contacts and hydrophobic interactions, reported for both the generated as well as the redocked pose.

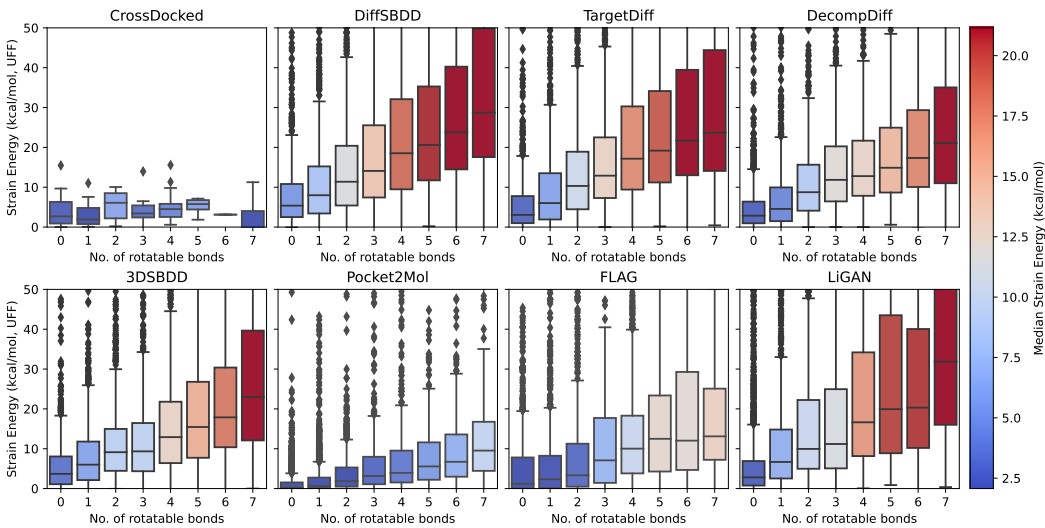

Figure 7: Boxplots of strain energies of generated molecules per number of rotatable bonds for all methods. Box color shows median strain value.

## C.2 STRAIN ENERGY

The additional analysis of the impact of rotatable bonds on strain and the frequency of rotatable bonds are shown in Figure 7 and Figure 8 respectively.

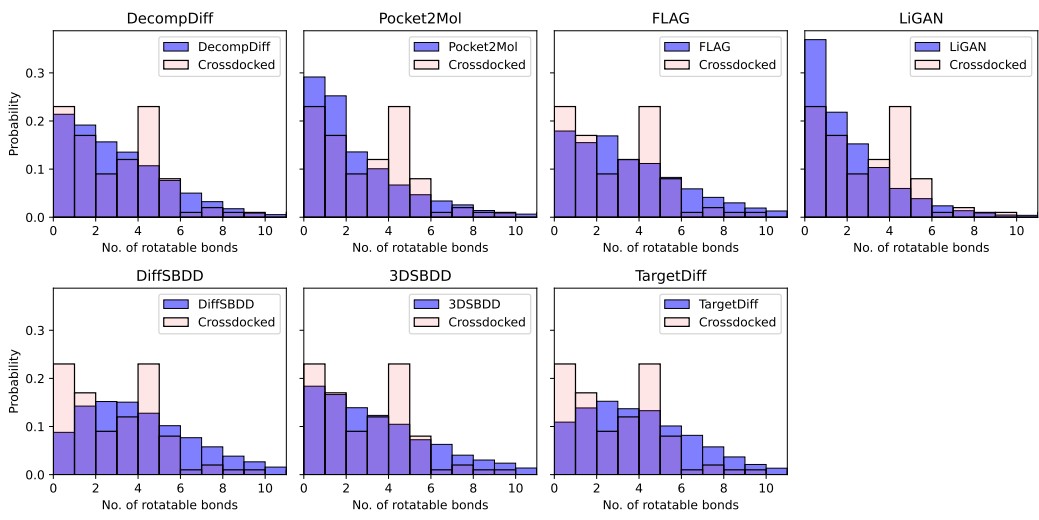

Figure 8: Histograms showing distributions of rotatable bonds for all molecules generated by a particular method. In each plot, the underlying distribution from CrossDocked is also shown.

