# OpenReview forum: "PoseCheck: Generative Models for 3D Structure-based Drug Design Produce Unrealistic Poses"
_ICLR.cc/2025/Conference — Submitted to ICLR 2025_

### Official Review · Reviewer_beLS · 2024-10-16

**Soundness:** 3
**Presentation:** 4
**Contribution:** 3
**Rating:** 5
**Confidence:** 5

**Summary:**

This paper proposed PoseCheck to directly evaluate the quality of molecular poses generated by 3D models. The authors introduced a number of metrics to SBDD, including interaction fingerprints, sterich clashes, strain energy, and redocking RMSD, and benchmarked the evaluation on CrossDocked dataset.

**Strengths:**

- PoseCheck highlights a key aspect in 3D molecule generation, i.e. the quality of generated poses as a prerequisite.
- This paper is generally well-written and easy-to-follow.
- The new metrics proposed faithfully evaluate the generated pose quality, sheding light on the fact that SBDD models might still have to cope with accurate pose modeling, which is informative to the SBDD community.

**Weaknesses:**

Major:
- Recent strong baselines are missing [1][2]. A more comprehensive evaluation would be needed for a higher score.
- I was wondering if the authors could provide a more indicative metric, similar to the PoseBusters passing rate [3]. Since metrics like steric clashes and strain energies are distributed within a certain range, it is not directly evident how those baseline models perform when they are not very significant outliers.

Minor:
- Citation format needs more careful handling. Misuses of \citet (\cite) are common in this paper which should be \citep instead.
- Line 653: by ”pocket similarity’ via Pocketome -> by ``pocket similarity'' via Pocketome

[1] Protein-Ligand Interaction Prior for Binding-aware 3D Molecule Diffusion Models. https://openreview.net/forum?id=qH9nrMNTIW

[2] MolCRAFT: Structure-Based Drug Design in Continuous Parameter Space. https://proceedings.mlr.press/v235/qu24a.html

[3] PoseBusters: AI-based docking methods fail to generate physically valid poses or generalise to novel sequences. https://pubs.rsc.org/en/content/articlehtml/2024/sc/d3sc04185a

**Questions:**

- Why is 3DSBDD and LiGAN even better than the CrossDocked dataset in terms of steric clashes? I feel that it does not make that much sense for generative models maximizing the data likelihood to surpass the dataset they've been trained on.
- What does the "a molecular weight ¡ 1000 Da" mean in Line 655?

---

### Official Review · Reviewer_RKP9 · 2024-11-02

**Soundness:** 2
**Presentation:** 2
**Contribution:** 1
**Rating:** 3
**Confidence:** 4

**Summary:**

The paper presents an approach to antibody design and optimization. However, the evaluation lacks comprehensiveness and fails to engage with several recent state-of-the-art methods in the field. This omission significantly impacts the robustness of the claims made regarding the effectiveness of the proposed method.

**Strengths:**

1. Innovative Concept: The paper introduces interesting concepts related to antibody design.
2. Potential Applications: The work has the potential for real-world applications in drug design, which is a valuable area of research.

**Weaknesses:**

1. As a benchmark, the evaluation in this article is far from comprehensive. Many recent works on antibody design and optimization are missing from the comparison, including GraphBP[1], VoxBind [2], MolCraft [3], and D3FG [4]. To the best of my knowledge, [2] and [3] are state-of-the-art methods, and all of the codes for these methods are open-sourced which I have tested. The lack of discussion and comparison with such a substantial body of related work is a major weakness.

2. Additionally, I believe the evaluation method in this article is far from comprehensive. First, aside from interactions, evaluating the molecular topology and structure in both 2D and 3D dimensions is essential. If the generated molecule is merely in a low-energy state (as described by DiffBP, where larger molecules can produce more interactions, thus lowering affinity) but significantly deviates from real drug data in terms of structure and chemical functional groups, can we truly consider that the generated molecule has an advantage?
Here, I have listed several methods and their examples in evaluating 3D geometric properties and 2D structural properties, refered to recently proposed benchmark paper [5], none of which have been considered in this article.
| Method       | Substructure       | Geometry                  |
|--------------|---------------------|---------------------------|
| LIGAN        | Figure. S6         | Figure. S7, S8, S9        |
| POCKET2MOL   | Table. 2           | Table. 3; Figure. 4       |
| GRAPHBP      | -                  | Table. 2; Figure. 5       |
| TARGETDIFF   | Table. 2           | Table. 1; Figure. 2       |
| DIFFBP       | Table. 3           | -                         |
| DIFFSBDD     | -                  | Figure. 8, 9              |
| FLAG         | Table. 3           | Table. 2; Figure. 4       |
| D3FG         | Table. 1, 3; Figure. 3 | Table. 2             |
| DECOMPDiff   | -                  | Table. 1, 2; Figure. 3    |
| MOLCRAFT     | Table. 1           | Table. 2                  |
| VOXBIND      | -                  | Figure. 7                 |

3. The conclusions in this paper also have significant issues. For example, in line 376, it states, “Interestingly, DiffSBDD and TargetDiff, which are considered state-of-the-art based on mean docking score evaluations.” However, DiffSBDD performs poorly in terms of the Vina score compared to other methods, indicating that its generated initial conformations are quite unstable. Yet, its final redocked energy is very low. Could this be due to the fact that the molecules generated by DiffSBDD have a higher molecular weight than those generated by other methods? If so, is this comparison truly fair? The Vina score is an essential metric for evaluating the quality of the generated initial conformations, and it should not be overlooked. Overall, the conclusions and analyses are overly simplistic and lack comprehensiveness.

4. The writing and presentation of this article are quite mediocre. For instance, the generation strategies, detailed introductions of methods include the model architecture and generative models, and classifications of these methods are not reflected in the tables, making the table arrangement appear rather arbitrary and unstructured.

In summary, I believe the evaluation in this article is incomplete, as it lacks many essential and reproducible methods. The conclusions are not sufficiently in-depth, and for existing methods, the article merely conducts a single test, followed by evaluations from various angles. In terms of both workload and quality, this paper falls short of the standards required for a high-level conference like ICLR. Therefore, I recommend rejection.

[1] Meng Liu, Youzhi Luo, Kanji Uchino, Koji Maruhashi, and Shuiwang Ji. Generating 3d molecules for target protein binding. ArXiv, abs/2204.09410, 2022.

[2] Pedro O. Pinheiro, Arian Jamasb, Omar Mahmood, Vishnu Sresht, and Saeed Saremi. Structure-based drug design by denoising voxel grids, 2024a.

[3] Yanru Qu, Keyue Qiu, Yuxuan Song, Jingjing Gong, Jiawei Han, Mingyue Zheng, Hao Zhou, and Wei-Ying Ma. Molcraft: Structure-based drug design in continuous parameter space. ICML 2024, 2024.

[4] Haitao Lin, Yufei Huang, Haotian Zhang, Lirong Wu, Siyuan Li, Zhiyuan Chen, and Stan Z. Li. Functional-group-based diffusion for pocket-specific molecule generation and elaboration. ArXiv, abs/2306.13769, 2023

[5] Haitao Lin, Guojiang Zhao, Odin Zhang, Yufei Huang, Lirong Wu, Zicheng Liu, Siyuan Li, Cheng Tan, Zhifeng Gao, Stan Z. Li, CBGBench: Fill in the Blank of Protein-Molecule Complex Binding Graph

**Questions:**

See Weaknesses.

---

### Official Review · Reviewer_Nu5A · 2024-11-03

**Soundness:** 3
**Presentation:** 3
**Contribution:** 2
**Rating:** 6
**Confidence:** 3

**Summary:**

This paper proposes new evaluation metrics for SBDD task, including redocked RMSD, steric clashes, interaction profile, and strain energy. Additionally, it conducts tests and analyses on several methods using these newly introduced metrics.

**Strengths:**

1. The current standard evaluation metrics for SBDD tasks are insufficient for adequately assessing the quality of models, which is a consensus within the community. This paper makes a valuable attempt to propose new evaluation metrics by incorporating insights from biophysical knowledge.
2. The evaluation and analysis of current methods highlight several issues present in existing deep learning approaches.

**Weaknesses:**

1. Although the proposed evaluation metrics have been comprehensively assessed on several existing methods, to my knowledge, some of the more advanced SBDD methods developed in the past two years have not been included.
2. Regarding interaction fingerprints, I agree that it is meaningful and interpretable to observe the interactions present in the generated complex conformations. However, the analysis and conclusions in this section are not sufficiently clear or thorough. For instance, I still do not understand how to evaluate a method in relation to the distribution of interaction counts. Is a higher number better, or is it more favorable for the distribution to be closer to that of the test set?

**Questions:**

1. In terms of agreement with docking scoring functions, is minimizing the RMSD with docking software considered optimal? As far as I know, docking software often has its own biases and errors. Is it possible that a method could actually yield conformations that are closer to the true structure but perform worse according to this metric?
2. Regarding the strain energy metric, should we only consider the strain energy of the individual molecule, or do we also need to account for the strain energy associated with conformational changes in the protein target?

**Details Of Ethics Concerns:**

NA.

---

### Official Review · Reviewer_Jsaw · 2024-11-04

**Soundness:** 3
**Presentation:** 3
**Contribution:** 3
**Rating:** 5
**Confidence:** 4

**Summary:**

This paper proposes a new set of metrics for evaluating SBDD molecular generation tasks. Four new metrics are introduced from the perspective of physical constraints, including redocked RMSD, steric clashes, interaction profile, and strain energy. In the experimental section, these metrics are tested on several important existing SBDD methods, and recommendations are provided based on the results.

**Strengths:**

The paper effectively highlights a critical issue in the current SBDD field, the evaluation metrics. The authors propose a two-tier approach to assessing SBDD-generated molecules: evaluating the intrinsic quality of the generated molecules and examining their structural conformations. For the latter, they introduce new metrics to address current limitations. While, from an application standpoint, the primary requirement is for SBDD models to produce effective small molecules capable of interacting with the binding pocket, supervising the quality of generated 3D conformations could offer valuable insights for further model refinement.

**Weaknesses:**

1. On the rationale of the strain energy metric: The energy change should ideally be observed as a whole, considering both the protein and the small molecule before and after binding, rather than focusing solely on the strain energy of the small molecule. "..., generally speaking, lower strain energy results in more favourable binding interactions and potentially more effective therapeutics." This assumption is inaccurate. When evaluating whether a protein and a small molecule are likely to bind and form a complex, the energy change of the protein cannot be neglected. Furthermore, as the authors pointed out, the generated poses are often problematic. The strain energy introduced by the authors may be influenced more by the generated pose itself than by the intended evaluation of the binding affinity or stability of the protein-ligand complex.
2. Regarding interaction fingerprinting, the authors present a distribution analysis of interaction fingerprints for several current models, noting significant deviations from the CrossDocked benchmark. However, the sheer number of hydrogen bond donors and acceptors does not determine the quality of the generated molecules. For instance, an excess of hydrogen bond donors and acceptors may reduce the selectivity of the small molecule, while selectivity is crucial in real-world pharmaceutical applications.
3. These metrics cannot directly guide drug discovery in real-world scenarios. Regarding redocked RMSD, the importance of this metric depends on how we define the function of SBDD models and what we expect for them. From a practical application perspective, the role of an SBDD model is  to provide potential candidate molecules, without necessarily ensuring the accuracy of their binding conformations.

**Questions:**

1. We know that the size of generated molecules has a significant impact on the docking score. For instance, in pocket2mol, as the sampled molecules increase in size, the Vina score tends to improve. It would be valuable to experimentally verify whether these new metrics are influenced by molecular size. Intuitively, the number of clashes is likely related to molecule size. A robust metric should ideally be unbiased concerning molecule size.
2. In the interaction analysis, it was observed that methods like diffsbdd, ligan, and decompdiff show a reduction in the number of hydrogen bond donors and acceptors after redocking. This is somewhat unexpected. What could be the possible reasons? Could this indicate that SBDD models are more sensitive to hydrogen bonding than redocking software?
3. Why does pocket2mol achieve similar performance to CrossDocked in terms of strain energy, despite not showing significant advantages in other metrics?What could be the possible reasons?

---

### Meta-Review · Area_Chair_QGHm · 2024-12-19

**Metareview:**

The paper addressed a significant challenge of unrealistic poses being generated by methods by AI applied to structure-based drug design. Reviewers raised significant issues that were not addressed during the rebuttal phase, including a critique of the introduced metrics.

Given the lack of further discussion during the rebuttal phase, I have to recommend rejection at this stage. Thank you for submitting your work for consideration to ICLR and I hope that the comments will be helpful in improving the manuscript.

**Additional Comments On Reviewer Discussion:**

The authors have not responded to comments and therefore I cannot provide comments on reviewers' discussion.

---

### Decision · Program_Chairs · 2025-01-22

Reject